

# A 12-years long (2010-2021) hydrological and biogeochemical dataset in the Sicily Channel (Mediterranean Sea)

Francesco Placenti[1], Marco Torri[2*], Katrin Schroeder[3], Mireno Borghini[4], Gabriella Cerrati[5], Angela Cuttitta[2], Vincenzo Tancredi[1], Carmelo Buscaino[1], Bernardo Patti[6]

[1]Consiglio Nazionale delle Ricerche - Istituto per lo studio degli impatti Antropici e Sostenibilità in ambiente marino (CNR-IAS), Campobello di Mazara (TP), Italy.
[2]Consiglio Nazionale delle Ricerche – Istituto di Studi sul Mediterraneo (CNR-ISMed), Palermo, Italy.
[3]Consiglio Nazionale delle Ricerche – Istituto di Scienze Marine (CNR-ISMAR), Venice, Italy.
[4]Consiglio Nazionale delle Ricerche – Istituto di Scienze Marine (CNR-ISMAR), La Spezia, Italy
[5]ENEA – Infrastrutture e Servizi – Servizio e Gestioni Centro Santa Teresa (ISER-STE), Santa Teresa, La Spezia, Italy;
[6]Consiglio Nazionale delle Ricerche - Istituto per lo studio degli impatti Antropici e Sostenibilità in ambiente marino (CNR-IAS), Palermo, Italy.

*Corresponding author: marco.torri@cnr.it

**ABSTRACT**

The data set presented here consists of 273 Conductivity-Temperature-Depth (CTD) stations, as well
as 2034 sampled data points in the water column, where dissolved inorganic nutrients have been
measured, that were collected during 12 summer oceanographic cruises (BANSIC series) in the Sicily
Channel (Central Mediterranean Sea), between 2010 and 2021. The quality of the CTD dataset is
ensured by regular sensor calibrations, an accurate control process adopted during the acquisition,
processing and post-processing phases. The quality of the biogeochemical dataset is ensured by the
adoption to best-practices analytical and sampling methods. This data collection fills up a gap of
information in the Sicily Channel, i.e. a key area where complex water mass exchange processes
involve the transfer of physical and biogeochemical properties between the Eastern and the
Western Mediterranean. The available dataset will be useful to evaluate the long-term variability
on a wide spatial scale, supporting studies on the evolution of the Mediterranean circulation and its
peculiar biogeochemistry, as well as on the physical and biogeochemical modeling of this area.
**INTRODUCTION**
The Mediterranean thermohaline circulation drives the transport of water masses and
biogeochemical elements in the different basins and sub-basins and, via the Strait of Gibraltar (SG),
controls the exchanges with the Atlantic Ocean (The MerMex Group, 2011).



The thermohaline circulation in the Mediterranean Sea (MS) is anti-estuarine and is mainly driven
by the balance between the relatively fresh waters entering at the SG and the negative fresh-water
budgets over the whole MS (Sorgente et al., 2011). Specifically, the Sicily Channel (SC), due to its
particular bathymetric structure and geographic position, plays a key role in modulating the
eastward transport of the fresher and superficial (0-150 m) Atlantic Water (AW) and the underlying
(200-500 m) westward transport of the salty Intermediate Water (IW) (Schroeder et al., 2017). From
its formation area (either in the Levantine sub-basin or in the Cretan Sea), IW spreads westward into
the Ionian Sea (IS), with a significant flow northward towards the Adriatic Sea, where it constitutes
an important preconditioning agent for the formation of the Adriatic Deep Water (ADW, which
forms the bulk of the Eastern Mediterranean Deep Water, or EMDW; e.g., Gačić et al., 2013). When
reaching the Western Mediterranean, and in particular its northern part, the IW preconditions the
water column also there and makes it prone to the formation of Western Mediterranean Deep
Water (WMDW; e.g., Roether et al., 1996). Although the area of the SC is limited at the east and
west by two relatively shallow sills (max depths of 350 m and 550 m, respectively), in its central part
the bottom depth can reach 1700 m. It is in this deep central trench where e.g., Gasparini et al.
(2005) and others studied the evolution of the upper part of the EMDW (or transitional EMDW, i.e.
tEMDW) over time, being able to cross the SC and reach the Tyrrhenian Sea along with the IW flow.
Several authors, analyzing long time series of temperature and salinity of the deeper waters of the
SC, have highlighted a general positive trend albeit characterized by phases of accelerations and
multiannual peaks and fluctuations (Gasparini et al., 2005; Gačić et al., 2013; Bonanno et al., 2014;
Ben Ismail et al., 2014; Schroeder et al., 2017; Placenti et al., 2022). Furthermore, these trends are
significantly faster than those reported for the global ocean intermediate layer (Borghini et al., 2014;
Schroeder et al., 2017). In fact, the semi-enclosed nature of the MS, together with its smaller inertia
due to the relative short residence time of its water masses, makes it highly reactive to external
forcings, identifying it as a ''hotspot'' for climate change (Giorgi, 2006). Consequently, MS is
expected to experience environmental impacts that are considerably greater than those in many
other places around the world (The MerMex Group, 2011). As regards the peaks, trends and
multiannual fluctuations of temperature and salinity observed in the deeper water of the SC, they
are probably ascribable to different processes acting at different spatial and temporal scales, such
as the passage of the signature of the Eastern Mediterranean Transient (EMT) (Gasparini et al.,
2005), the alternation of circulation phases (cyclonic-anticyclonic) of the Northern Ionian Gyre (NIG)



(Gačić et al., 2013; Bonanno et al., 2014; Placenti et al., 2022) and the effects related to warming of
the Eastern Mediterranean (Schroeder et al. al., 2017, 2019).
The anti-estuarine circulation, jointly to the superposition of different time scales of variability,
intense wintertime atmospheric forcings, NIG reversals and EMT, act also on the distribution of
biogeochemical elements (e.g., inorganic nutrients) and productivity of the MS. The very low
productivity of the MS is therefore mainly linked both to the anti-estuarine circulation (Krom et al.,
2010) and to the chemical speciation of the dissolved P and N. They in fact reflect a switch from less
bioavailable chemical forms of P and N entering the Mediterranean Sea to more bioavailable forms
leaving it (Powley et al., 2017). Moreover, the export of nutrients through the IW causes the deep
waters of the Eastern Mediterranean Sea to be more nutrient depleted than deep water in all other
parts of the global ocean (Krom et al., 2005). Another peculiarity still debated is the higher molar
$NO_3:PO_4$ ratio in the deeper water of the MS compared to the "classical" world oceans Redfield
ratio, indicating a general P-limited regime, which becomes stronger along a west-to-east gradient
(Belgacem et al., 2020). In this context, the aim of this paper is to compile a large dataset of
hydrological (temperature, salinity and pressure) and biogeochemical (nitrate, phosphate and
silicate) properties from *in situ* data collected between 2010 and 2021 in the SC, filling up a gap of
information in a key area of the MS, where the exchange between the two basins is taking place.
The available dataset is a valuable tool in support to the evaluation of the long-term variability and
evolution of the Mediterranean circulation and water masses, and provides also a useful
contribution for the implementation of models aimed at describing the physical-chemical processes
occurring in this area. The dataset could also be integrated in the recently published climatology of
dissolved inorganic nutrients (Belgacem et al., 2020), to expand its geographic domain.

**DATASET AND METHODS**
The hydrological and biogeochemical data collection was carried out as part of an ichthyoplankton
monitoring and research program implemented by the Italian National Research Council (CNR). This
program was set up with the primary purpose of monitoring the spatio-temporal distribution of the
early life stages of European anchovy (*Engraulis encrasicolus*) in the SC and of studying its
relationship with the environmental variables, in support to the sustainable exploitation of the
population by local fisheries. In this context, since 1998 ichthyoplankton oceanographic surveys
based on a common sampling grid have been annually carried out during the summer period within
the FAO Geographic Sub-Areas (GSAs) 13, 15, 16 and 19. Since 2010, in addition to the meso-

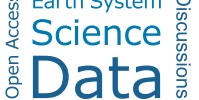

zooplancton sampling, water sampling for the quantification of macro-nutrients has been included
in the survey work plan in order to better characterize the biogeochemical characteristics of the
water column and study the relationships with the biotic component.
The dataset presented here results from this sampling effort and assembles information from 12
summer oceanographic cruises conducted on board of different research vessels from 2010 to 2021
(R/V "Urania" from 2010 to 2014; R/V "Minerva Uno" from 2015 to 2017; R/V "G. Dallaporta" from
2018 to 2021). Data were integrated into a dataset consisting of 273 CTD-nutrient stations and 2034
data points (Suppl. 1). The stations are arranged along inshore-offshore transects approximately sub
perpendicular to the Sicilian coast, aiming at characterizing the oceanographic and biogeochemical
features in a key area for the understanding of the complex exchange processes between the
Eastern and Western basins (Fig. 1).

*Figure 1. Stations map of the Bansic cruises, carried out in the Sicily Strait from 2010 to 2021: CTD*

*and nutrient stations are indicated by blue (in the general map) and red circles (in the yearly maps).*

*The maps were created using Ocean Data View software (https://odv.awi.de/).*



**Hydrological Data Acquisition**

At all stations, pressure, salinity, and temperature were measured with a CTD (conductivity, temperature, and depth) probe (Sea-Bird Scientific) mod. SBE 911plus and a General Oceanics rosette with 24 Niskin bottles of 12 L capacity. Temperature measurements were performed with a SBE-3/F thermometer, with a resolution of 0.00015 °C/bit at -1 °C or 0.00018 °C/bit at 31 °C, and conductivity measurements were performed with a SBE-4C sensor, with a resolution of 3 x 10-4 S/m. The vertical profiles of all parameters were obtained by sampling the signals at 24 Hz, with the CTD/rosette going down at a speed of 1 m/s. The rosette is equipped with a sonar altimeter which intercept the bottom 100-70 meters before getting to it. The altimeter is used just for safety, to avoid the rosette to touch the bottom.

**Inorganic Nutrient Data Collection**

Seawater samples for dissolved inorganic nutrient analysis were collected from the surface to the bottom by means of Niskin bottles. In particular, during the CTD upcast, a variable number of water samples, at selected standard depth, has been considered (surface–25m–50m–75m–100m–150m–200m–300m–400m–500m–600m–700m–800m–900m–1000m–bottom) with slight modifications in the upper layer where significant hydrological variability is typical to occur. All materials used for water sampling on board were earlier conditioned with 10% HCl and rinsed 3 times with ultrapure water. Unfiltered samples were stored on board at –20°C.

**Analytical Methods for Inorganic Nutrients**

For all cruises, nutrient determination (nitrate, silicate, and phosphate) was carried out following standard colorimetric methods of seawater analysis, defined by Grasshoff et al. (1999) and Hansen and Koroleff (1999) adapted to an automated system. Specifically, the determination of phosphate is based on the colorimetric method, in which a blue color is formed by the reaction of phosphate, molybdate ion and antimony ion, followed by reduction with ascorbic acid. The reduced blue phospho-molybdenum complex is read at 880 nm. Inorganic nitrate is reduced to nitrite at pH 8 in a copperized cadmium reduction coil that reacts with an aromatic amine, leading to the final formation of the azo dye measured at 550 nm. Then, the nitrite that is separately determined must be subtracted from the total amount measured to get the nitrate concentration only. The determination of soluble silicates is based on the reduction of a silico-molybdate complex in acid solution to molybdenum blue by ascorbic acid and the absorbance is measured at 820 nm.



All the analysis of dissolved inorganic nutrients were carried out immediately after each
oceanographic cruise, in the nutrient laboratory of the Institute for the Study of Anthropic Impacts
and Sustainability in the Marine Environment (CNR-IAS) of Capo Granitola, using the same analytical
instrument and the same scientific staff. The concentration (µmol/l) of nitrate, silicate and
phosphate was measured by means of a Sial Autoanalyzer ''QUAATRO''. The detection limits for
nitrates, silicates and phosphates were 0.02, 0.01 and 0.006 µmol/l, respectively. Even though the
use of the same analytical methods, instruments and scientific staff supports the repeatability and
the comparison of the measurements, in order to  further validate  the analytical data, selected
seawater samples (sampled in duplicate) have been sent to the nutrient laboratory of nutrients of
the Research Center (ENEA) of Santa Teresa (La Spezia), taking advantage on their participation in
the framework of the European intercalibration program QUASIMEME (Quality Assurance of
Information for Marine Environmental Monitoring in Europe).
The differences in concentrations for all parameters analyzed (nitrates, phosphates and silicates)
ranged from 3% to 20%. The differences were greater (10-20%) for concentration values close to
the instrumental detection limit and smaller (<10%) at high concentrations. This range of differences
is perfectly acceptable considering that ENEA uses a previous generation auto-analyser and that the
scientific staff was different. However, we would like to point out that, in both nutrient laboratories,
the chemical analyzes were carried out using both the same analytical methods and the same types
of reagents.

**Quality check of hydrological and nutrient data**
The temperature and salinity of the CTD have been regularly calibrated. During 2 cruises, also
redundant temperature and salinity sensors were used. When they were available, the secondary
sensors have been used to assess the stability of the primary ones.
The temperature and salinity sensors calibrations have been performed before each cruise by CNR
technicians at the NATO Centre for Underwater Research (NURC, now Centre for Maritime Research
and Experimentation, CMRE) in La Spezia (Italy) until 2016. Between 2017 and 2018 sensors were
send to the manufacturer, while since 2019 the calibration is done at the new CNR-ISMAR
calibration laboratory in La Spezia (Italy). Table 1 shows a summary of all sensors, their serial
numbers and their calibration dates.

*Table 1 – Calibration dates and serial numbers of the CTD sensors used during oceanographic cruises.*



| Cruise | Date | Temp 1 | | Cond 1 | | Temp 2 | | Cond 2 | |
|---|---|---|---|---|---|---|---|---|---|
| | | sn | cal. date | sn | cal. date | sn | cal. date | sn | cal. date |
| Bansic 2010 | 25 Jun–14 Jul 2010 | 1368 | May10 | 891 | May10 | | | | |
| Bansic 2011 | 08–26 Jul 2011 | 4440 | Apr11 | 3172 | Apr11 | | | | |
| Bansic 2012 | 04–23 Jul 2012 | 1183 | Nov10 | 923 | Nov10 | | | | |
| Bansic 2013 | 26 Jun–16 Jul 2013 | 2810 | Oct12 | 2483 | Oct12 | | | | |
| Bansic 2014 | 22 Jul–9 Aug 2014 | 4440 | Nov13 | 3172 | Nov13 | | | | |
| Bansic 2015 | 16 Jul–3 Aug 2015 | 5022 | Oct14 | 3485 | Nov14 | | | | |
| Bansic 2016 | 30 Jun–14 Jul 2016 | 5022 | Oct14 | 3485 | Nov14 | | | | |
| Bansic 2017 | 13–29 Jun 2017 | 1183 | Jul16 | 0923 | Jun16 | | | | |
| Bansic 2018 | 07-19 Sep 2018 | 1142 | Aug17 | 2779 | Aug17 | | | | |
| Bansic 2019 | 30 Sep-12 Oct 2019 | 1142 | Aug17 | 2779 | Feb19 | 5038 | May17 | 3484 | Feb19 |
| Bansic 2020 | 16-25 Sep 2020 | 1142 | Jan20 | 2779 | Jan20 | 5038 | Jan20 | 3484 | Jan20 |
| Bansic 2021 | 6-18 Sep 2021 | 1381 | Jun21 | 1048 | Jun21 | | | | |


After their acquisition, CTD data were pre-processed by the SBE Data Processing™ software, in order
to (i) convert the raw data (.hex) to engineering units and store them in a .cnv file, (ii) run a low-pass
filter on the data and smooth high frequency data, (iii) align parameter data in time, relative to
pressure (to ensure that calculations of salinity and other parameters are made using
measurements from the same parcel of water), (iv) remove conductivity cell thermal mass effects
from the measured conductivity, (v) compute derived variables, and to (vi) average data, using
averaging intervals based on depth range, and split the file into an upcast and a downcast file.
Following the recommendations of the SeaDataNet QC guidelines (SeaDataNet, 2010) The
subsequent procedure to assess data quality was based on the following list:
-    Check header details (vessel, cruise number, station numbers, date/time, latitude/longitude

171         (start and end), instrument number and type, station depth, cast (up or down)), data

172         type/no. of data points)

-    Plot station positions to check not on land
-    Check ship speed between stations to look for incorrect position or date/time
-    Automatic range checking of each parameter
-    Check units of parameters supplied
-    Check pressure increasing
-    Check no data points below bottom depth
-    Plot profiles (individually, in groups, etc)
-    Check for spikes
-    Check for vertical stability/inversions
-    Plot temperature vs. salinity
The resulting dataset is based on the downcast file after selection of averaged data at standard
depths corresponding to the water sampling for the inorganic nutrient analysis.



Furthermore, a further control process was carried out taking into account the characteristics of the
three water masses identified in the study area, related to 2010-2021 period, and schematized
below: a surface layer with a thickness varying over the years, generally less than 150 m of depth
and mainly occupied by AW, an intermediate layer (200-500 m) mainly occupied by IW and a deeper
layer (>500 m) occupied by the upper part of the DW (Fig. 2h). The first step consisted in the
elimination of the outliers from nitrates, phosphates and silicates profiles recorded during annual
surveys in each layer. Then, the mean value and the coefficient of variation (CV), i.e., a normalized
measure of the dispersion given by the ratio of the standard deviation to the mean, were calculated
for each parameter recorded, with the aim of carrying out a comparative control of the occurring
patterns.

**RESULTS AND DISCUSSION**
An analysis was conducted in order to characterize the spatial-temporal trends emerging in the
dataset and compare them with the state of the art concerning the study area. In this framework,
the vertical distribution pattern of inorganic nutrients in the water column of the SC highlights low
concentration values and high variability (Fig. 2; Tab. 2).

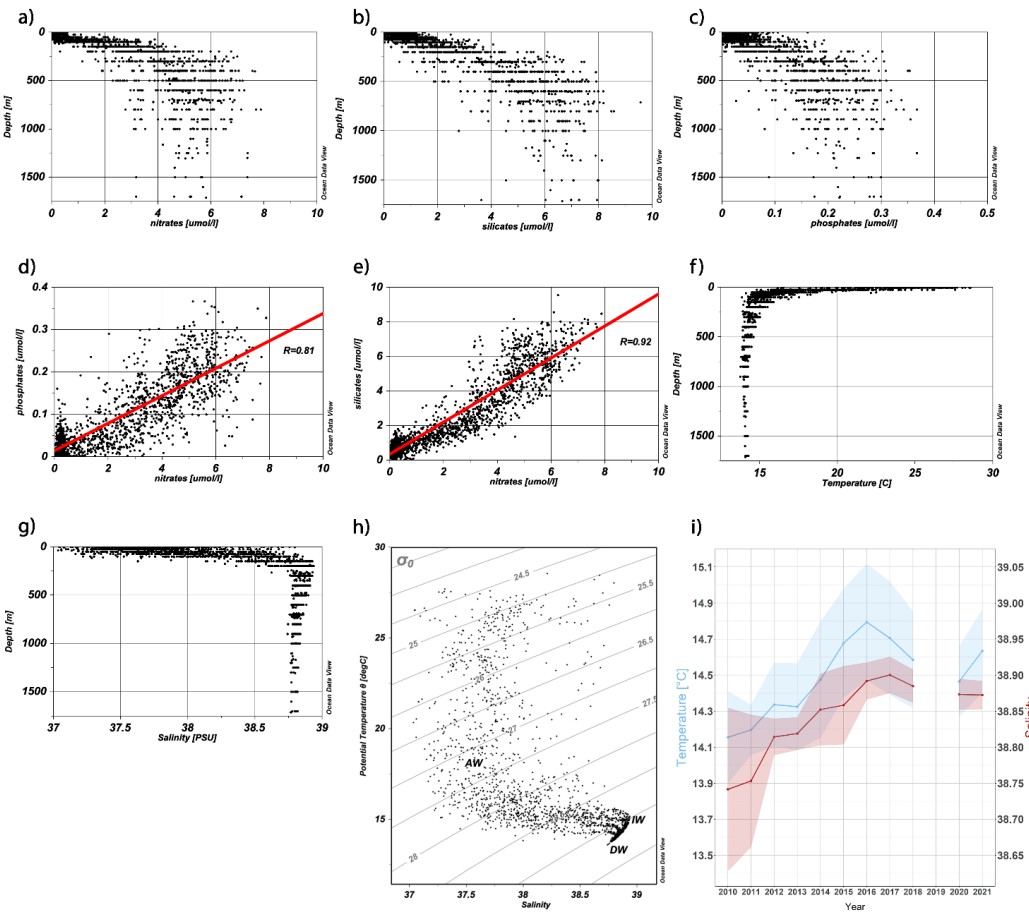


*Figure 2. Plots of data on nutrients concentration at selected standard depths along the water column for a) nitrates, b) phosphates, c) silicates; d) N:P and e) N:Si diagrams and related linear regression lines (in red); f) plots of hydrological data at selected standard depths along the water column for temperature and g) salinity (the related colored areas represent the standard deviations); h)potential temperature vs salinity diagram related to 2010-2021 period (AW for Atlantic Water, IW for Intermediate Water and DW for upper Deep Water) and i) time series of annual average values (2010-2021) of temperature (blue line) and salinity (red line) related to the water sampling depths in the IW (200-500 m).*


In surface waters the concentration values of inorganic nutrients are close to the instrumental detection limits due to the typical consumption of phytoplankton during the summer period occurring in this layer. This usually results in considerably lower values compared to patterns



observed in the underlying layers (Tab. 2). Moreover, the depth layer 0-150m is characterized by a
higher CV due to the pronounced dynamism of the exchange processes affecting the marine
ecosystem in this upper part of the water column as well as the interaction with terrestrial sources
of nutrients. Specifically, the lowest mean nitrate concentration value of 0.302 µmol/l was
measured in BANSIC19 survey, characterized by a quite low number of water samples collected from
continental shelf stations only (no data of the intermediate layer are available from 2019 survey),
while the highest concentration value of 1.027 µmol/l was measured in BANSIC14 survey (Tab. 2).
Regarding the phosphates the lowest mean concentration value was 0.030 µmol/l (BANSIC12) and
the highest one 0.064 µmol/l (BANSIC13), while for silicates the mean concentration values ranged
from a minimum of 0.608 µmol/l (BANSIC15) to a maximum of 1.173 µmol/l (BANSIC10) (Tab. 2).
The lower sampling effort carried out in BANSIC19 also corresponds to a lower variation of nitrates
and phosphates compared to the other surveys, while the silicates showed more homogeneous
patterns among surveys. Specifically, in the superficial layer CV varied between 0.762 (BANSIC19)
and 1.566 (BANSIC20) for nitrates, between 0.471 (BANSIC19) and 1.333 (BANSIC21) for phosphates,
and finally between 0.433 (BANSIC20) and 0.83 (BANSIC15) for silicates.
The nutrients that have been consumed at the surface are regenerated in the mesopelagic layer by
bacteria and animals (due to respiration), increasing the nutrient concentrations in the deeper water
masses over time (e.g., Schroeder et al., 2010). In this way, the intermediate waters of the SC are
characterized by higher nutrient concentration values than the overlying layer and lower variability
(Fig. 2a-c and Tab. 2). Specifically, mean concentration values in the intermediate waters ranged
from 2.976 µmol/l in BANSIC15 to 5.4 µmol/l in BANSIC11 for nitrates, from 2.589 µmol/l (BANSIC15)
to 4.86 µmol/l (BANSIC11) for silicates, and from 0.095 µmol/l in BANSIC18 to 0.204 µmol/l in 2021
for phosphates (Tab. 2).
Regarding the dispersion of the values in this layer, nitrates CV ranged between 0.0068 (BANSIC20)
and 0.319 (BANSIC21), phosphates CV ranged between 0.171 (BANSIC20) and 0.585 (BANSIC16),
and silicates CV ranged between 0.161 (BANSIC20) and 0.528 (BANSIC15) (Tab. 2).
In the trench of the SC, the deep layer (see section >500m-bottom of Tab. 2) is characterized by
mean concentration values more homogeneous over years than in the above layers. There, the
lowest mean concentration values for nitrates and silicates (4.13 and 4.701 µmol/l) were measured
in BANSIC15, while the lowest value for phosphates (0.097 µmol/l) was measured during BANSIC10.
The highest mean values of nitrates (5.861 µmol/l), phosphates (0.241 µmol/l) and silicates (6.474
µmol/l) were respectively measured during BANSIC11, BANSIC17-21 and BANSIC13. Similarly, the



deep layer is characterized by the lowest variability in nutrient concentrations. Indeed, the nitrate
CV varied between 0.0058 (BANSIC10) and 0.299 (BANSIC21), between 0.07 (BANSIC20) and 0.339
(BANSIC10) for the phosphates and between 0.029 (BANSIC20) and 0.277 (BANSIC15) for the
silicates.
The molar ratios of nitrate:phosphate (N:P) and silicate:nitrate (Si:N) in the whole water column of
the SS show high correlation coefficients  (R=0.81 and 0.92, respectively) (Fig. 2d,e). Specifically, in
the surface layer (0-150m) the mean values of N:P and Si:N over the period 2010-2021 are equal to
18 and 1.4 respectively, which become 31 and 0.9 in the intermediate layer (200-500m) and  29 and
1.1 in the deep layer (500m-bottom), in agreement with Ribera D'Alcalà et al. (2003) and with
Placenti et al. (2013; 2022).  Observed N:P values are higher than the classical Redfield ratio (16:1).
The high N:P ratio can result from either a decrease in phosphate or an increase in nitrate; however,
the reason of this anomaly is still unclear (e.g., Schroeder et al., 2010). Among the various
hypotheses, we agree about the possible role of external inputs, together with very limited
denitrification (Krom et al., 2010; Huertas et al., 2012; Van Cappellen et al., 2014), in explaining the
observed very high $NO_3:PO_4$ ratios in the deeper water (Powley et al., 2017).

*Table 2. Average values of concentration and coefficients of variation (CV) of both physical*
*(temperature and salinity) and chemical (nitrates, phosphates and silicates) parameters in the 12*
*oceanographic surveys carried out yearly over the period 2010-2021 reported in this paper, by depth*
*layer (superficial 0-150m, intermediate 200-500m and deep >500m-bottom).*

| Cruise name | Sampling date | Research vessel name | Station number | Max. Sampling depth [m] | Sample number | Nitrates mean [umol/l] | Nitrates CV | Phosphates mean [umol/l] | Phosphates CV | Silicates mean [umol/l] | Silicates CV | Temperature mean [°C] | Temperature CV | Salinity mean [psu] | Salinity CV |
|---|---|---|---|---|---|---|---|---|---|---|---|---|---|---|---|
| | | | | | | | | **Layer 0-150 m** | | | | | | | |
| Bansic10 | 25 June–14 Jul 2010 | Urania | 13 | 700 | 62 | 0.743 | 1.172 | 0.043 | 1.009 | 1.173 | 0.697 | 17.477 | 0.148 | 37.610 | 0.011 |
| Bansic11 | 08–26 Jul 2011 | Urania | 34 | 1130 | 154 | 0.736 | 1.446 | 0.034 | 0.831 | 1.002 | 0.686 | 18.574 | 0.234 | 37.978 | 0.0087 |
| Bansic12 | 04–23 Jul 2012 | Urania | 39 | 1711 | 203 | 0.707 | 1.066 | 0.030 | 0.954 | 1.037 | 0.478 | 17.923 | 0.237 | 38.177 | 0.0117 |
| Bansic13 | 26 June–16 Jul 2013 | Urania | 34 | 1700 | 178 | 0.835 | 1.302 | 0.064 | 0.727 | 1.061 | 0.628 | 17.380 | 0.175 | 37.919 | 0.0094 |
| Bansic14 | 22 Jul–9 Aug 2014 | Urania | 32 | 1700 | 164 | 1.027 | 1.116 | 0.051 | 0.803 | 0.940 | 0.703 | 17.997 | 0.201 | 37.872 | 0.0123 |
| Bansic15 | 16 Jul–3 Aug 2015 | Minerva Uno | 31 | 1700 | 158 | 0.681 | 1.117 | 0.036 | 0.893 | 0.608 | 0.833 | 18.313 | 0.240 | 38.206 | 0.0093 |
| Bansic16 | 30 June–14 Jul 2016 | Minerva Uno | 10 | 1700 | 51 | 0.777 | 1.237 | 0.043 | 0.638 | 0.614 | 0.690 | 17.673 | 0.176 | 37.812 | 0.0135 |
| Bansic17 | 13–29 June 2017 | Minerva Uno | 32 | 1700 | 160 | 0.898 | 1.015 | 0.037 | 1.037 | 1.043 | 0.538 | 17.880 | 0.179 | 37.956 | 0.0125 |
| Bansic18 | 07-19 Sept 2018 | G. Dallaporta | 16 | 700 | 77 | 0.957 | 1.286 | 0.039 | 0.671 | 1.150 | 0.649 | 18.646 | 0.229 | 38.091 | 0.0118 |
| Bansic19 | 30 Sept-12 Oct 2019 | G. Dallaporta | 6 | 72 | 21 | 0.302 | 0.762 | 0.040 | 0.471 | 0.721 | 0.605 | 19.784 | 0.189 | 38.037 | 0.0068 |
| Bansic20 | 16-25 Sept 2020 | G. Dallaporta | 11 | 1000 | 44 | 0.596 | 1.566 | 0.032 | 0.697 | 1.083 | 0.433 | 19.861 | 0.222 | 38.091 | 0.0096 |



| | | | | | | | | | | | | | | | |
|---|---|---|---|---|---|---|---|---|---|---|---|---|---|---|---|
| Bansic21 | 6-18 Sept 2021 | G. Dallaporta | 15 | 996 | 67 | 0.577 | 1.449 | 0.073 | 1.333 | 1.087 | 0.526 | 19.623 | 0.222 | 37.822 | 0.0128 |
| **Layer 200-500 m** | | | | | | | | | | | | | | | |
| Bansic10 | 25 June–14 Jul 2010 | Urania | 13 | 700 | 26 | 4.086 | 0.238 | 0.166 | 0.445 | 4.606 | 0.309 | 14.155 | 0.018 | 38.742 | 0.0029 |
| Bansic11 | 08–26 Jul 2011 | Urania | 34 | 1130 | 65 | 5.400 | 0.273 | 0.135 | 0.443 | 4.860 | 0.364 | 14.196 | 0.010 | 38.753 | 0.0024 |
| Bansic12 | 04–23 Jul 2012 | Urania | 39 | 1711 | 68 | 4.407 | 0.244 | 0.170 | 0.342 | 4.443 | 0.342 | 14.336 | 0.016 | 38.814 | 0.0007 |
| Bansic13 | 26 June–16 Jul 2013 | Urania | 34 | 1700 | 65 | 4.473 | 0.216 | 0.163 | 0.338 | 4.623 | 0.271 | 14.325 | 0.017 | 38.819 | 0.0006 |
| Bansic14 | 22 Jul–9 Aug 2014 | Urania | 32 | 1700 | 61 | 4.753 | 0.194 | 0.178 | 0.298 | 4.047 | 0.307 | 14.474 | 0.022 | 38.852 | 0.0013 |
| Bansic15 | 16 Jul–3 Aug 2015 | Minerva Uno | 31 | 1700 | 57 | 2.976 | 0.402 | 0.148 | 0.476 | 2.589 | 0.528 | 14.677 | 0.020 | 38.858 | 0.0014 |
| Bansic16 | 30 June–14 Jul 2016 | Minerva Uno | 10 | 1700 | 23 | 4.002 | 0.277 | 0.096 | 0.585 | 3.097 | 0.416 | 14.794 | 0.022 | 38.892 | 0.0007 |
| Bansic17 | 13–29 June 2017 | Minerva Uno | 32 | 1700 | 52 | 4.017 | 0.290 | 0.167 | 0.397 | 4.193 | 0.294 | 14.705 | 0.021 | 38.900 | 0.0006 |
| Bansic18 | 07-19 Sept 2018 | G. Dallaporta | 16 | 700 | 16 | 4.460 | 0.142 | 0.095 | 0.520 | 4.223 | 0.185 | 14.584 | 0.018 | 38.885 | 0.0006 |
| Bansic19 | 30 Sept-12 Oct 2019 | G. Dallaporta | 6 | 72 | | | | | | | | | | | |
| Bansic20 | 16-25 Sept 2020 | G. Dallaporta | 11 | 1000 | 11 | 4.301 | 0.068 | 0.115 | 0.171 | 3.869 | 0.161 | 14.464 | 0.013 | 38.873 | 0.0006 |
| Bansic21 | 6-18 Sept 2021 | G. Dallaporta | 15 | 996 | 13 | 5.025 | 0.319 | 0.204 | 0.304 | 4.677 | 0.254 | 14.634 | 0.016 | 38.872 | 0.0005 |
| **Layer >500-bottom** | | | | | | | | | | | | | | | |
| Bansic10 | 25 June–14 Jul 2010 | Urania | 13 | 700 | 4 | 4.420 | 0.006 | 0.097 | 0.339 | 4.987 | 0.045 | 13.908 | 0.004 | 38.777 | 0.0002 |
| Bansic11 | 08–26 Jul 2011 | Urania | 34 | 1130 | 25 | 5.861 | 0.125 | 0.179 | 0.280 | 6.392 | 0.163 | 13.946 | 0.003 | 38.787 | 0.0002 |
| Bansic12 | 04–23 Jul 2012 | Urania | 39 | 1711 | 49 | 5.413 | 0.171 | 0.216 | 0.217 | 6.468 | 0.163 | 13.997 | 0.005 | 38.780 | 0.0003 |
| Bansic13 | 26 June–16 Jul 2013 | Urania | 34 | 1700 | 39 | 4.886 | 0.158 | 0.207 | 0.212 | 6.474 | 0.150 | 14.019 | 0.003 | 38.794 | 0.0002 |
| Bansic14 | 22 Jul–9 Aug 2014 | Urania | 32 | 1700 | 37 | 5.546 | 0.106 | 0.239 | 0.190 | 6.088 | 0.127 | 14.069 | 0.003 | 38.811 | 0.0003 |
| Bansic15 | 16 Jul–3 Aug 2015 | Minerva Uno | 31 | 1700 | 26 | 4.130 | 0.240 | 0.238 | 0.267 | 4.701 | 0.277 | 14.125 | 0.003 | 38.817 | 0.0003 |
| Bansic16 | 30 June–14 Jul 2016 | Minerva Uno | 10 | 1700 | 16 | 5.774 | 0.240 | 0.118 | 0.339 | 6.180 | 0.237 | 14.222 | 0.006 | 38.834 | 0.0005 |
| Bansic17 | 13–29 June 2017 | Minerva Uno | 32 | 1700 | 25 | 4.954 | 0.231 | 0.241 | 0.203 | 6.205 | 0.117 | 14.229 | 0.003 | 38.837 | 0.0003 |
| Bansic18 | 07-19 Sept 2018 | G. Dallaporta | 16 | 700 | 3 | 5.315 | 0.036 | 0.159 | 0.093 | 5.905 | 0.084 | 14.200 | 0.003 | 38.832 | 0.0003 |
| Bansic19 | 30 Sept-12 Oct 2019 | G. Dallaporta | 6 | 72 | | | | | | | | | | | |
| Bansic20 | 16-25 Sept 2020 | G. Dallaporta | 11 | 1000 | 7 | 4.650 | 0.017 | 0.155 | 0.070 | 5.038 | 0.030 | 14.201 | 0.003 | 38.834 | 0.0002 |
| Bansic21 | 6-18 Sept 2021 | G. Dallaporta | 15 | 996 | 7 | 5.474 | 0.299 | 0.241 | 0.279 | 6.229 | 0.191 | 14.248 | 0.003 | 38.815 | 0.0002 |


As far as the hydrological properties of the water masses in the study area, it is worth noting that
over the (summer) period 2010-2021 in the surface layer the average temperature varied in the
range 17.380-19.861 °C, with a general increasing trend peaking in 2020. Similarly, the salinity
showed increasing values in the range 37.610-38.206 with a peak of in 2015 (Tab. 2). This layer is
characterized by a strong interannual and spatial variability as subject to continuous interaction with
the atmosphere, as also highlighted by one or two higher orders of magnitude in CV values of
physical parameters in the superficial layer compared to the deeper layers (Fig. 2f, g; Tab. 2). Indeed,
in the upper layer the CV of temperature varied between 0.148 (BANSIC10) and 0.240 (BANSIC15),
and the CV of salinity ranged between 0.007 (BANSIC19) and 0.014 (BANSIC16).
Instead, the deeper water masses, with their lower variability in heat and salt content turn out to
be much more suitable for monitoring any changes over long periods. In the IW layer the CV of
temperature ranged between 0.013 (BANSIC20) and 0.022 (BANSIC14 and BANSIC16), while the CV
of salinity ranged between 0.0005 (BANSIC21) and 0.0029 (BANSIC10) (Tab. 2). Values of
temperature in our time series (2010-2021) shows an increase in average values of about 0.5 °C
(14.155-14.634 °C), with a peak of 14.794 in 2017 (Tab. 2). Similarly, the corresponding increase in
salinity was about 1.15 psu (37.742-38.885) with a peak of 38.900 psu reached in 2018 (Tab. 2).
Similar trends have been also highlighted in the same area by Schroeder et al. (2017) and related to
the increase to drying processes affecting the surface waters from which the LIW originates. Within
the time series (2010-2021) two patterns can be distinguished, the first (2010-2016) is characterized
by an average annual increase in temperature (dT/dt) of about 0.1 °C/year, the second one (2017-
2020) by an annual decrease in temperature of 0.08 °C/year (Fig. 2i; Tab. 2). Finally, in 2021 a slight
rise in temperature occurred (Fig. 2i; Tab. 2). Similarly, the salinity patterns show an average annual
increase (dS/dt) of 0.022 over the period 2010-2016 and an annual decrease of 0.007 afterwards
(2017-2021) (Fig. 2i; Tab. 2). A recent study advanced hypotheses on a possible link between the
temperature and salinity positive trends over the period 2010-2016 and the anticyclonic phase of
the NIG over the years 2006-2010, while the negative trends over the period 2017-2021 have been
connected to the cyclonic phase of the NIG during years 2011-2016 (Placenti et al., 2022).
Conversely, the nitrates and silicates mean annual concentrations show a slight decrease up to
2015-2016 and then slight increase until 2021 (Tab. 2), in agreement with the state of the art
regarding the SS (Placenti et al., 2022).
The DW are characterized by very low values in CV of both temperature and salinity (between 0.003-
0.006 and 0.002-0.005, respectively) with very slight differences across the yearly surveys (Tab. 2).
Moreover, the average annual temperature and salinity slightly increased from 2010 until 2017
(13.908-14.299 °C and 38.777-38.837), and then slightly decreased until 2020 (Tab. 2). On the
contrary, the patterns of nitrates and silicates would appear to be characterized by a slight decrease
(5.861-4.130 and 6.392-4.701 µmol/l) between 2011 and 2015 and by a slight discontinuous
increase afterwards until 2021 (Tab. 2).

**DATA AVAILABILITY**



Dataset and metadata are available as a *.csv merged file and *xlsx format respectively from
ZENODO and will be accessed in full open access form at https://doi.org/10.5281/zenodo.8125006
(Placenti et al., 2023) after acceptance.

**CONCLUSION**
The dataset described here has the advantage of having been collected within the same monitoring
program implemented on an important area of the Mediterranean Sea. Over time, the data has
been collected consistently by the same technicians and researchers, who have been able to apply
a rigorous quality check during both field and laboratory activities. This made it possible to minimize
any biases deriving from the different experience of the personnel and from the use of different
types of instrumentation and protocols. The spatial and temporal variability that characterizes this
dataset therefore reliably reflects the effect of environmental processes that occurred in the marine
environment, making this dataset usable for different fields of application. In support to this, the
analysis of the mean values and of the variability of the parameters along the water column and
among different surveys showed an agreement with the patterns commonly recognized in the
marine environment and highlighted in other datasets collected in different areas of the
Mediterranean Sea. Moreover, the analysis shown in this paper is highly consistent with the trends
evidenced in the literature concerning the Sicily Channel. In this framework, the availability of this
dataset to the scientific community fills an important lack in field observations of a crucial area, the
Sicily Channel, where exchanges between the western and eastern Mediterranean basin take place,
providing support to studies aimed at describing the ongoing processes as well as at realizing reliable
projections regarding the effects of these processes in the near future.

**AUTHOR CONTRIBUTIONS**
FP wrote the manuscript and analyzed the dataset. MT wrote the manuscript and prepared the
figures. VT and CB participated to the fieldwork and laboratory analyses. KS wrote the manuscript
and improved the quality check of the oceanographic data. GC, AC and BP coordinated the field
work and the laboratory analyses and supervised the writing of the text. MB managed the technical
aspects related to the oceanographic instrumentation for the acquisition of the hydrological
parameters and helped with specific technical aspects of the manuscript. All authors contributed to
the article and approved the submitted version.



**COMPETING INTERESTS**
The authors declare that they have no conflict of interest.

**FINANCIAL SUPPORT**
This study was mainly supported by the Italian National Research Council (CNR) through USPO office
and by the FAO Regional Project MedSudMed "Assessment and Monitoring of the Fishery Resources
and the Ecosystems in the Straits of Sicily", co-funded by the Italian Ministry MIPAAF through the
Directorate General for Maritime Affairs and Fisheries of the European Commission (DG MARE).
Other national research programmes supported this study including project SSD-PESCA,
coordinated by the Ministry of the Education, University and Research (MIUR) and founded by the
Ministry of Economic Development (MISE), and the Flagship Project RITMARE – The Italian Research
for the Sea, coordinated by the Italian National Research Council and funded by MIUR.

**ACKNOWLEDGMENTS**
Masters of the Urania, Minerva Uno, Dallaporta and all their crew are thanked for their work in
support to the sampling activities during the oceanographic cruises. We are grateful to Carmelo
Bennici, Girolama Biondo, Gaspare Buffa, Ignazio Fontana, Giovanni Giacalone, Luigi Giaramita,
Marianna Musco, Carlo Patti and Giorgio Tranchida for their valuable technical support and the
sampling collection during the oceanographic surveys.

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
