# Peer review of "A 12-years long (2010-2021) hydrological and biogeochemical dataset in the Sicily Channel (Mediterranean Sea)"

_Earth System Science Data, 2023_

## Referee Comment (RC1)

This work presents a data set made of temperature, salinity, nitrate, phosphate and silicate concentrations at certain depth levels (0, 25 50, 75, 100, 200, etc…) at certain locations of the Sicily Channel.

The manuscript is well written and is clear. The dataset is really interesting and provides a very interesting information that will be made freely available after the publication of the present manuscript. For all these reason I find that it is suitable for publication in Earth Science Data after the correction of some minor questions.

Detailed comments:

The abstract is clear and reflects what the reader is going to find in this work.

the introduction is well written and focuses the reader on the subject of this work. There are only some minor questions.

Lines 15-17 does not seem clear to me.

Lines 18-20. "The thermohaline circulation is not driven only by the balance between the fresh water entering at the Strait of Gibraltar and the negative freshwater budget over the whole Mediterranean". The thermohaline circulation is driven by the negative freshwater budget and the net heat loss over the whole Mediterranean. The entrance of freshwater through the Strait of Gibraltar is the consequence of these two deficits: freshwater and heat. On the other hand, the freshwater balance is not only between the freshwater entering through Gibraltar and the negative budget. The exit of salty Mediterranean water has the same importance in this balance. Beside this, the exit of Mediterranean water is also driven by the deficit of freshwater and heat.

Lines 23 and 24 is the first time you mention Intermediate Waters (IW). This term is used throughout all the manuscript. Obviously you refer to IW originated in the Levantine Basin, basically Levantine Intermediate Water and Cretan Intermediate Water. With this redaction, the reader could think that these are the only intermediate water masses in the Mediterranean Sea. There is also an intermediate water originated in the Western Mediterranean. Somehow you have to clarify this in this manuscript.

Dataset and methods.

Line 80. You mention several GSAs. It would be interesting to know in which GSA is the Sicily Channel.

Hydrological adquisition.

The value I would like to know is the accuracy in ºC and in salinity units. When I see a temperature, I want to know on which decimal place I can trust and which one is not accurate. The resolution, in ºC/bit gives no information to me.

Analythical methods.

According to Line 124, I understand that nitrites are also measured. Why this dataset only provides the information concerning nitrates. Why not providing also nitrites?

Quality check…

Line 188. Once again IW, without indicating its Levantine origin.

Line 190. Elimination of outliers. Please, specify how outliers are defined. $>2.5\sigma$, or $>3\sigma$, or whatever… ($\sigma$ is standard deviation).

Lines 191 to 194. When I read these lines I do not understand how you have calculated the coefficients of variation. Then I have understood it reading the rest of the manuscript and looking at table II. But it should be clarified here. If $\mu$ is the mean value and $\sigma$ the standard deviation, $C=\sigma/\mu$. These mean values and standard deviations are calculated for each depth level (0, 25, 50, 75, 100, 150, 200, 300, etc.) using the data of all the oceanographic stations and all the campaigns? Or they are calculated for each campaign at each depth level using the data of all the stations? Looking at table II I understand that you calculate the mean values for all the stations and for all the depth levels within a certain depth range for the upper layer, then for a certain depth range for the intermediate layer, and the same for the deep layer. Is this right?

Figure 2. In the legend it is stated that b) is phosphates, but the x-axis says silicates. Most importantly. I cannot see the axis without a loupe. Please use larger letters and numbers.

Lines 229-230. Nutrients are regenerated in the mesopelagic layer by bacteria and animals (due to respiration). I am not a specialist, but I would only talk about bacteria. As you are using nitrate, I think that nitrification is carried out by bacteria. Animals can excrete ammonium, that can be used by some phytoplanktonic groups, but if I think about nitrate, the bacteria are the organisms responsible for oxidation of ammonium to nitrite first, and then to nitrate. As I said, I am not sure, but saying that animals regenerate nutrients sounds strange to me.

Line 251. SS. I suppose that you mean SC for Sicily Channel. SS has not been defined in the manuscript.

---

## Author Response (AR1)

*We thank the two referees for the useful comments that allowed us to improve the quality of our work. We have positively received all the referees' suggestions. Below, we present point-by-point responses to the comments.*

**REFEREE 1:** This work presents a data set made of temperature, salinity, nitrate, phosphate and silicate concentrations at certain depth levels (0, 25 50, 75, 100, 200, etc...) at certain locations of the Sicily Channel. The manuscript is well written and is clear. The dataset is really interesting and provides a very interesting information that will be made freely available after the publication of the present manuscript. For all these reason I find that it is suitable for publication in Earth Science Data after the correction of some minor questions. Detailed comments: The abstract is clear and reflects what the reader is going to find in this work. the introduction is well written and focuses the reader on the subject of this work. There are only some minor questions. Lines 15-17 does not seem clear to me. Lines 18-20. "The thermohaline circulation is not driven only by the balance between the fresh water entering at the Strait of Gibraltar and the negative freshwater budget over the whole Mediterranean". The thermohaline circulation is driven by the negative freshwater budget and the net heat loss over the whole Mediterranean. The entrance of freshwater through the Strait of Gibraltar is the consequence of these two deficits: freshwater and heat. On the other hand, the freshwater balance is not only between the freshwater entering through Gibraltar and the negative budget. The exit of salty Mediterranean water has the same importance in this balance. Beside this, the exit of Mediterranean water is also driven by the deficit of freshwater and heat.

*We modified this part in agreement with the reviewer suggestions, highlighting the contribution of the net heat loss in the circulation of the Mediterranean Sea (lines 15-19)*

**REFEREE 1**: Lines 23 and 24 is the first time you mention Intermediate Waters (IW). This term is used throughout all the manuscript. Obviously you refer to IW originated in the Levantine Basin, basically Levantine Intermediate Water and Cretan Intermediate Water. With this redaction, the reader could think that these are the only intermediate water masses in the Mediterranean Sea. There is also an intermediate water originated in the Western Mediterranean. Somehow you have to clarify this in this manuscript.

*As we are interested in the intermediate waters flowing in the Strait of Sicily, it is now better specified that the IW mentioned in the manuscript are referred to the intermediate waters originating in the Eastern Mediterranean Basin, as the Levantine Intermediate Water and Cretan Intermediate Water (lines 20-25)*

**REFEREE 1**: Dataset and methods. Line 80. You mention several GSAs. It would be interesting to know in which GSA is the Sicily Channel.

*We now better specified that the GSAs 13, 15, 16 are all located in the Strait of Sicily (lines 80-81).*

**REFEREE 1**: Hydrological adquisition. The value I would like to know is the accuracy in °C and in salinity units. When I see a temperature, I want to know on which decimal place I can trust and which one is not accurate. The resolution, in °C/bit gives no information to me.

*We met the reviewer's suggestion by adding accuracy information related to hydrological data (lines 101-104).*

**REFEREE 1**: Analythical methods. According to Line 124, I understand that nitrites are also measured. Why this dataset only provides the information concerning nitrates. Why not providing also nitrites?

*In order to align the new hydrological and biogeochemical information of the Strait of Sicily with dataset already present in other areas of the Mediterranean (e.g. Belgacem et al 2020), we preferred to include only nitrates, silicates and phosphates in the dataset. In addition, we preferred to exclude nitrites as they did not provide additional information to our discussion.*

**REFEREE 1**: Quality check... Line 188. Once again IW, without indicating its Levantine origin.

*As already mentioned in a previous comment, it is now better specified that the IW mentioned in the manuscript are referred to the intermediate waters originating in the Eastern Mediterranean Basin (lines 20-25).*

**REFEREE 1**: Line 190. Elimination of outliers. Please, specify how outliers are defined. >2.5s, or >3s, or whatever… (s is standard deviation). Lines 191 to 194. When I read these lines I do not understand how you have calculated the coefficients of variation. Then I have understood it reading the rest of the manuscript and looking at table II. But it should be clarified here. If μ is the mean value and s the standard deviation, C=s/μ. These mean values and standard deviations are calculated for each depth level (0, 25, 50, 75, 100, 150, 200, 300, etc.) using the data of all the oceanographic stations and all the campaigns? Or they are calculated for each campaign at each depth level using the data of all the stations? Looking at table II I understand that you calculate the mean values for all the stations and for all the depth levels within a certain depth range for the upper layer,then for a certain depth range for the intermediate layer, and the same for the deep layer. Is this right?

*The reviewer is right. Indeed, the different depth level (0, 25, 50, 75, 100, 150, 200, 300, etc.) were used in order to characterize the profile of the parameters in the water column in each sampled station. Then, in order to characterize the three identified water masses in each annual survey, we calculated the mean and the CV of the all values recorded in the same water mass for each survey, as reported in the Table 2. In the lines 185-200 it is now specified in more detail the methodology followed for the identification of the outliers and for the calculation of the mean and the coefficient of variation.*

**REFEREE 1**: Figure 2. In the legend it is stated that b) is phosphates, but the x-axis says silicates. Most importantly. I cannot see the axis without a loupe. Please use larger letters and numbers.

*We modified the caption on the figure 2 according to the reviewer suggestion.*

**REFEREE 1**: Lines 229-230. Nutrients are regenerated in the mesopelagic layer by bacteria and animals (due to respiration). I am not a specialist, but I would only talk about bacteria. As you are using nitrate, I think that nitrification is carried out by bacteria. Animals can excrete ammonium, that can be used by some phytoplanktonic groups, but if I think about nitrate, the bacteria are the organisms responsible for oxidation of ammonium to nitrite first, and then to nitrate. As I said, I am not sure, but saying that animals regenerate nutrients sounds strange to me.

*Following this suggestion, the sentence has been modified (lines 236-238).*

**REFEREE 1**: Line 251. SS. I suppose that you mean SC for Sicily Channel. SS has not been defined in the manuscript.

*We thank the reviewer. We corrected adopting the name "Sicily Channel (SC)" in the all part of the manuscript.*

**REFEREE 2:** This paper describes a dataset of temperature, salinity and inorganic nutrient measurements collected during various cruises in the Sicilian Channel. The article is well-written, and clear and represents an important source of data that can be used for further studies. I have only minor comments to suggest to the authors. General comments: Although the article is well written, it is difficult to read due to too many acronyms, which sometimes force one to search the text already read. I recommend significantly reducing the number of acronyms, leaving only the most used ones.

*The number of acronyms has been reduced according to the comment (e.g. SG, ADW, EMDW, WMDW)*

**REFEREE 2:** Specific comments: r25-28: give some reference depths;

*The sentence has been modified.*

**REFEREE 2:** r40-42: semi-enclosed basins have higher residence time than open basins. This sentence does not seem correct. Specify it better. Specify also if you are speaking of lateral forcing (Gibraltar Strait) or surface forcing (atmosphere);

*The sense of this sentence is that the Mediterranean Sea have a shorter residence time (about 70 years) compared to the other oceans (200-1000 years), as reported in e.g. The MerMex Group et al. (2011). Therefore, following the reviewer comment, the sentence has been modified by adding more information (lines 41-44).*

**REFEREE 2:** r162: Specify the file type not only the extension (cnv);

*The sentence has been modified accordingly (line 164).*

**REFEREE 2:** r191: I have some doubts about the use of CV. For example, it is not well defined if the mean is 0 (tide). However, this is not the case. Have you some reference works?

*We calculated the CV following the approach used by Belgacem et al. 2020 (cited in the manuscript). Moreover the methodological part has been now modified by adding more detailed information (lines 186-200).*

**REFEREE 2:** fig 2: the text in the parenthesis should stay in panel i) not g) probably;

*The sentence has been modified accordingly (lines 215-216).*

**REFEREE 2:** r284: Acronym is not defined;

*The acronym has been corrected (line 296).*

**REFEREE 2:** r297: Acronym is not defined;

*The acronym has been corrected (line 200).*

**REFEREE 2:** r306: Also here please specify the file type not only the extension.

*The sentence has been modified (lines 318-319).*